# Evaluating Spatiotemporal Distribution of Residential Sprawl and Influencing Factors Based on Multi-Dimensional Measurement and GeoDetector Modelling

**DOI:** 10.3390/ijerph18168619

**Published:** 2021-08-15

**Authors:** Linlin Zhang, Guanghui Qiao, Huiling Huang, Yang Chen, Jiaojiao Luo

**Affiliations:** 1School of Tourism and Urban-Rural Planning, Zhejiang Gongshang University, Hangzhou 310018, China; zhanglinlin@mail.zjgsu.edu.cn; 2School of Architecture and Civil Engineer, Heilongjiang University of Science and Technology, Harbin 150022, China; somnusjay@gwu.edu; 3Law School, Ningbo University, Ningbo 315211, China; chenyang2@nbu.edu.cn; 4School of Economics, Zhejiang University of Finance & Economics, Hangzhou 310018, China; luojiaojiao@zufe.edu.cn

**Keywords:** residential sprawl, multi-dimensional measurement, CO_2_ emissions, incremental planning, GeoDetector modeling

## Abstract

Residential sprawl constitutes a main part of urban sprawl which poses a threat to the inhabitant environment and public health. The purpose of this article is to measure the residential sprawl at a micro-scale using a case study of Hangzhou city. An integrated sprawl index on each 1 km × 1 km residential land cell was calculated based on multi-dimensional indices of morphology, population density, land-use composition, and accessibility, followed by a dynamic assessment of residential sprawl. Furthermore, the method of GeoDetector modeling was applied to investigate the potential effects of location, urbanization, land market, and planning policy on the spatial variation of residential sprawl. The results revealed a positive correlation between CO_2_ emissions and residential sprawl in Hangzhou. There has been a remarkable increase of sprawl index on residential land cells across the inner suburb and outer suburb, and more than three-fifths of the residential growth during 2000–2010 were evaluated as dynamic sprawl. The rapid development of the land market and urbanization were noted to impact the spatiotemporal distribution of residential sprawl, as *q*-statistic values of population growth and land price ranked highest. Most notably, the increasing *q*-statistic values of urban planning and its significant interactions with other factors highlighted the effects of incremental planning policies. The study derived the policy implication that it is necessary to transform the traditional theory and methods of incremental planning.

## 1. Introduction

The 2018 Revision of World Urbanization Prospects predicted that the proportion of the world’s population living in urban areas will increase to 68% by 2050. Nearly 90% of future increases in the world’s urban population are expected to be highly concentrated in Asia and Africa. Especially, China’s unrivaled urbanization has drawn worldwide attention [1,2]. How to sustain the increasing urban residents in a resource-saving pattern, therefore, become a focal issue of sustainable development for Chinese cities. However, urban sprawl which is defined in terms of “undesirable” land-use patterns [3], has gradually been a prevailing phenomenon in large cities across the country, such as Beijing [4], Guangzhou [5], Shanghai [6], Nanjing [7], Chongqing [8] and Hangzhou [9], etc. As noted by scholars, urban sprawl in China mainly consists of leapfrogged industrial parks [10] and low-density residential communities that are discontinuous from existing urban centers [8]. Along with the boom of building new towns and development zones, urban residential development has become one of the dominant growth patterns in suburbs since the mid-1990s [11,12]. From 2006 to 2015, China witnessed an explosive expansion in residential land at a growth rate of 66.62%, which accounted for 37.38% of the incremental urban construction land. It can be foreseen that residential sprawl should be a key theme of future research into urban China.

From a global perspective, urban sprawl has been widely criticized for aggravating a series of urban and environmental problems [13,14,15], such as increasing energy consumption [16] and emissions of air pollutants [17,18], intensifying urban heat islands [19], lowering the quality or quantity of ecosystem services [20], obesity and chronic disease [21,22], etc. In spite of the potential costs of sprawl, there is little consensus on valid and reliable measures of urban sprawl, due to the ambiguity in its definition. Without a universally accepted definition, urban sprawl seems distinctive at various scales and to different researchers [3]. The mainstream consensus is that sprawl should be interpreted from multiple dimensions. Based on a systematic comparison of definitions, the most important connotations of urban sprawl were captured (Figure 1).

Morphology: urban areas expand fragmentally and dispersedly [23]. Urban growth can be distinguished into three types: infilling, edge expansion, and leaping development [9,24]. Among them, leaping development (discontinues development) consumes more space resources than continuous development [25].Density: a low-density situation of urban activities [3,26,27], which can be described as built units per unit of developable land [28], urban construction land per capita, or the number of settlements/residents/employment per unit area [29].Composition: the pattern of land development is homogenous, characterized by poor diversity of mixed land-uses [7]. Particularly, a high percentage of a certain single land-use indicates that it is homogenous and thus sprawling [29].Accessibility: poor accessibility of related land uses to one another [3], meaning that residents and workers have to commute large distances to reach a destination [30]. Accessibility can be expressed by road lengths, travel time, cost, or distance [3,29].

Relating to diverse interpretations of urban sprawl, a variety of approaches have been utilized to quantify it, without a general agreement [15]. The previous quantification approaches can be grouped into two categories, namely single-dimensional measurement, and multi-dimensional measurement (Table 1). For single-dimensional measurement [14,31], density is the most popular sprawl measure [26], which is perhaps the most important dimension of sprawl [14,28]. Besides, whether urban land expanding faster than population growth or not matters, as low-density development leads to overconsumption of land resources. The ratio between the growth rate of built-up areas and the population growth rate is frequently used [14,27,32] by characterizing urban growth as “sprawling” if the quotient is higher than 1 [29]. However, later multi-dimensional methods proved the insufficiency of single-dimensional measurement which could not fully capture urban sprawl [28,32]. Numerous multi-dimensional measurements have been put forward [26,29,30,32,33,34,35,36,37], based on different combinations of indicators from the dimensions of density, configuration, composition, accessibility, etc. Several studies used the approach combining single-dimensional measurements with multi-dimensional measurements [32] to generate more reliable and robust results. In addition, existing measurements of urban sprawl differ greatly in temporal and spatial scales, and most of them were on macro-scale [14,31] or meso-scale [26,27,34]. As Wilson et al. [38] noted, it is necessary to capture the relative intensity of sprawl at different times or various areas. For example, Zeng et al. [36] proposed an integrated multi-level and multi-dimensional method to characterize urban sprawl, and their specifications of levels were parcel at micro-level, district at meso-level, and metropolitan area at macro-level. Comparatively, inadequate research has been conducted on micro-scale with the smallest unit of intra-city blocks or units [35], which compared the variations of urban sprawl across different regions or different time periods [9]. Such kind of micro-scale studies on intra-city grids will be of significance for more differential and precise responses to urban growth.

The elaborated literature is mainly looking at the total urban land, without discriminating one certain land-use from the others. Although sprawl has many aspects, its residential aspect is central to its status and potential effects, and other aspects of sprawl are dependent on and driven by how and where people live to a certain extent [14]. Few empirical studies put focus on the urban sprawl on residential land-use, except Galster et al. [26] and Crawford [39]. What’s more, the transitional context emphasizes the importance of interpretative research on residential sprawl in China. Liu et al. [40] assessed the determinants of suburban residential development using accessibility factors and neighborhood factors, with no reference to government factors in its statistic models. Later, Liu et al. [8] attributed China’s urban sprawl to failures in government and market forces under the land financial incentives. Wu et al. [41] investigated the driving forces of suburban residential landscapes in Beijing by using a framework of growth coalition which assumed that local government and real estate enterprises achieve lower land acquisition costs and higher profits for housing. The literature offered a rich analytical perspective in analyzing the issue in a transitional economy though they did not directly correspond to the phenomenon of residential sprawl [11,42]. However, research efforts have seldom been spent on quantitatively assessing the influencing factors for residential sprawl. Overall, studies on the spatiotemporal evolution and explanatory factors of residential sprawl will expand the understanding of Chinese urban growth.

This paper aims to build a multi-dimensional measurement approach, and an empirical case study will be conducted to measure the temporal and spatial dynamic characteristics of residential sprawl in Hangzhou City. By using the method of GeoDetector modeling, the influencing factors for the variation of the residential sprawl index will be quantitatively examined. Indubitably, it will be of significance to fill the literature gaps, not only for enriching multi-dimensional measurements of intra-urban residential sprawl at the micro-scale but also for a better understanding of the formation mechanism of residential sprawl in a transitional country.

## 2. Materials and Methods

### 2.1. Multi-Dimensional Measurement 

#### 2.1.1. Sprawl Index

On the basis of the literature, urban sprawl is a complex phenomenon that cannot be measured by only one or two measures [29]. Instead, the multi-dimensional connotation of sprawl requires various measures for each dimension [13,26]. Scholars have asserted that characterizations of sprawl are not similar across the world [26,28,30], which poses a challenge for the generalizability of measurements. Seeking to account for the unique characteristics of regions, a few multi-dimensional measurements even confounded the causes and consequences of urban sprawl with the phenomenon itself, which was widely criticized [13,15,26]. Findings from different multi-dimensional measurements usually cannot be compared with each other and, therefore, may be difficult to interpret consistently [15]. After all, an ideal measuring approach should be applicable as well as duplicable nationally or internationally [14]. Based on the principle, this research proposed a multi-dimensional measuring approach that captures the cardinal dimensions of sprawl: morphological discontinuity, low density, low mixed-use degree of land-use composition, and poor accessibility. The details of the four dimensions were already described above, referring to Figure 1. A complete explanation of all indices used in this study and the direction of their impact on the degree of sprawl are presented in Table 2. 

Because of the disunity of magnitudes and value range, the indices should be standardized so that they can be comparable to each other. The calculation formula is as follows: (1)$$\mathrm{for}\;\mathrm{positive}\;\mathrm{indices},\;\;\;\;x_{ij}^{\prime }=\frac{\left(x_{ij}-x_{jmin}\right)}{\left(x_{jmax}-x_{jmin}\right)}$$
(2)$$\mathrm{for}\;\mathrm{negative}\;\mathrm{indices},\;\;\;\;x_{ij}^{\prime }=\frac{\left(x_{jmax}-x_{ij}\right)}{\left(x_{jmax}-x_{jmin}\right)}$$
where *x_ij_* is the value of index j of unit *i*, *x_jmax_* and *x_jmin_* is the maximum and minimum of index *j* in the whole area respectively.

The average weighted comprehensive sprawl index is calculated as follows:(3)$$SI_{i}=\overset{n}{\underset{j=1}{\sum }}w_{f}\times X_{ij}^{\prime }$$
where $X_{ij}^{\prime }$ is the standardized value of the index *j* of unit *i*, *w_f_* is the weight coefficient of index *j*. The larger $SI_{i}$ is, the higher level of urban sprawl, and vice versa.

Finally, the sprawl index on residential land was extracted by utilizing an overlay analysis of the layers of comprehensive urban sprawl index and residential land.

#### 2.1.2. Dynamic Assessment

The spatiotemporal scaling effect of urban sprawl is an essential basis for the accurate diagnosis of urban sprawl. Two dynamic indices, therefore, were used to characterize the spatiotemporal attributes of urban sprawl. The growth ratio of residential land expansion to population growth was calculated by the following formula [32]:(4)$$\;GR=\frac{A_{t}-A_{0}}{A_{0}}-\frac{P_{t}-P_{0}}{P_{0}}$$
where *A_0_* and *A_t_* refer to the residential land area at the beginning and the end respectively, *P_0_* and *P_t_* refer to the population at the beginning and the end respectively. GR>0 means that residential land is consumed at a significantly faster rate than population growth, which indicates a situation of inefficient residential growth, and vice versa.

The urban growth types were distinguished using the following equation [24]:(5)$$S=\frac{L_{c}\;}{P}$$
where $L_{c}$ is the length of the common boundary of an incremental construction land patch and existing urban construction land patch, and P is the perimeter of this newly developed residential patch. When *S* ≥ 0.5, the growth type is identified as infilling; when 0 < *S <* 0.5, it is identified as edge expansion; and when *S* = 0, it is identified as leaping with no common boundary.

### 2.2. GeoDetector Modelling

The method of GeoDetector modeling proposed by Wang et al. [44] has been increasingly used to detect spatial differentiation and identify the driving factors [45,46,47]. It assumes that if an independent variable X has an important influence on a dependent variable Y, then the spatial distribution of X and Y should be similar [47]. In this study, two sub-modules were adopted: factor detector and interaction detector. 

#### 2.2.1. Factor Detector

The factor detector was used to detect the spatial heterogeneity of urban sprawl index Y and how a certain factor X explains the spatial patterns of urban sprawl index Y. The degree of explanation is measured by the *q*-statistic [44], which is denoted as: (6)$$q=1-\frac{1}{N\sigma^{2}}\overset{L}{\underset{h=1}{\sum }}N_{h}\sigma_{h}^{2}$$
where *h* = 1, ..., *L* is the stratification of variable Y or factor X; the study area is composed of *N* units, $N_{h}$ is the number of units in layer h; $\sigma_{h}^{2}$ and $\sigma^{2}$ are the variances of the Y value of layer h and the whole area respectively. The range of *q* is [0,1], a higher value of *q* indicates a stronger spatially stratified heterogeneity of the dependent variable Y and a stronger explanatory power of factor X to the attribute Y, and vice versa.

#### 2.2.2. Interaction Detector

The interaction detector was applied to assess whether the explanatory powers of two factors are enhanced, weakened, or independent of each other. The *q* values of two factors and that of their interaction would be calculated [46]. The new layer of interaction is formed by the tangent of overlay variables X1 and X2. By comparing *q*(X1) and *q*(X2) with *q*(X1∩X2), the interaction type between two factors can be identified [45]: the interaction relationship between two factors is bivariate enhancement when max(*q*(X1), *q*(X2)) < *q*(X1∩X2) < *q*(X1) + *q*(X2), while that is non-linear enhancement if *q*(X1∩X2) > *q*(X1) + *q*(X2). 

#### 2.2.3. Variables

Based on the literature [6,8,40,42], this study proposed to detect major factors that influence the variation of the residential sprawl index (the dependent variable) by using explanatory variables in Table 3. The locational variables of West Lake (X1) and Qiantang river (X2) not only reflect the effect of the natural environment but may also imply the impacts of Hangzhou’s urban spatial strategy “From the West Lake era to the Qiantang River era”. The variable of population growth (X3) is supposed to represent the urbanization factor. The land cost variables reflecting the factor of the land market consist of the benchmark price of residential land (X4) and the transaction price of residential land (X5). The factor of urban plan policy (X6) indicates how the cell is distant from urban centers and new towns.

### 2.3. Study Area

Hangzhou is a metropolis of East China and the capital of Zhejiang Province, which is located at 29°11′–30°34′ N and 118°20′–120°37′ E. The city has experienced rapid urbanization, with its urbanization rate changed from 29.40% in 1990 to 67.38% in 2019. Meanwhile, the urban built-up area increased from 69 km^2^ to 648.46 km^2^, which implies that its land urbanization has far outpaced the population urbanization. As Hangzhou witnessed several changes in the boundary of urban districts, the study area (Figure 2) was based on eight urban districts in 2010 with an area of 3068 km^2^, when and where the database is comparable and available to obtain. It is necessary to define the division of urban core and suburban area, where the urban core is the built-up area in pre-1949, the inner suburb includes the area outside of the urban core but within the main city, and the outer suburb consists of two Yuhang district and Xiaoshan district (refer to Zhang et al. [10]).

## 3. Results

### 3.1. Multi-Dimensional Sprawl Index on Residential Land Cells

Based on the land-use datasets, the area of urban residential land in the study area saw an increase of nearly 80 km^2^ during the decade 2000–2010, ranked as the second-largest proportion of urban land growth after industrial land. Compared to the relatively compact pattern before 2000, residential land dispersed outward tremendously (Figure 3). The most residential growth emerged in the west of the urban area, where a batch of residential areas was assembled, mainly in streets or towns of Jiangcun, Wuchang, Xianlin, and Yuhang. Apart from this, there were increasing residential agglomerations in the northwest (Sandun and Liangzhu) and northeast (Jiubao and Linping new town) of the city during the decade. Moreover, it is noteworthy that the residential development has dispersed toward the southeast and across the Qiantang River, i.e., Binjiang and Xiaoshan. The area of residential land within the inner suburb grew by 1.7 from 2000 to 2010 but did not carry a higher proportion of total urban residential land, conversely, decreasing from 49.61% to 39.88%. At the same phase, a much steeper rise of residential areas emerged in the outer suburb from 16.27 km^2^ to 57.61 km^2^. The outer suburb hereby replaced the inner suburb as the most expansive area, with its residential expansion accounted for 55.52% of the total residential growth.

The spatial distribution and historical changes of sprawl index on 1 km × 1 km urban construction land cells are shown in Figure 4b,d. The values of sprawl indexes (SI) varied in a wide range both in 2000 and 2010: among the cells on urban construction land, 13.31%~14.06% cells were identified as not sprawl (SI < 0.5), 46.81%~47.59% cells were identified as mild sprawl (0.5 ≤ SI < 0.65), and 39.1% cells were identified as severe sprawl (0.65 ≤ SI < 1). The sprawl degree of the cells within the urban core was the lowest, the farther away from the center, the higher the sprawl index was. From 2000 to 2010, the average sprawl index decreased in the urban core (from 0.40 to 0.35) and the inner suburb (from 0.6 to 0.56). Whereas, the average sprawl index of cells in the outer suburb increased greatly, from 0.62 in 2000 to 0.63 in 2010.

For the terms of each dimension, as shown in Figure 4a,c, the variation of shape index, population density, mixed-use degree, and road density corroborated an aggravating sprawling tendency in the city. For example, the average population density of cells within the urban core increased from 23,768 persons per km^2^ to 30,405 persons per km^2^, whereas that of cells in the suburban area decreased from 2890 persons per km^2^ to 1691 persons per km^2^, which indicates that there was a huge density gap between the urban center and the suburban area. Cells within the urban core increased the average mixed-use level significantly from 0.46 to 0.62, and so was the mixed-use degree of the inner suburb, which increased from 0.25 to 0.34. On the contrary, the outer suburb saw a decrease in the mixed-use degree (from 0.21 to 0.19), where industrial land and residential land-uses dominated. Besides the old residential areas in the urban core, most residential land-use cells were distributed in suburban new towns. Similarly, the variation accessibility level also saw a decreasing law from the city center to the fringe. Overall, the urban sprawl within the inner suburb dominated before 2000, while the outer suburb far outweighed in the later period.

Figure 4e,f demonstrates the spatial layout of SI on residential land cells of 2000 and 2010, respectively. The spatial distribution of SI underwent significant changes in the past years, among which the most obvious changes were in the outer suburb, followed by the inner suburb. In 2000, there were about 21.24 km^2^ of residential land identified as mild or severe sprawl (SI ≥ 0.5), accounting for 52.08% of the total residential land area. The sprawling residential land cells of 2000 mainly gathered around Binjiang Newtown, and other sporadic sprawling residential cells scattered in Xiaoshan district and Yuhang district. In 2010, the sprawling residential land further expanded toward the urban fringe, which covered 59.11 km^2^ of mild sprawl (0.5 ≤ SI < 0.65) and 7.81 km^2^ of severe sprawl (0.65 ≤ SI < 1), accounting for 58.06% of the total residential land area. In 2000 and 2010, the residential land without sprawl (SI < 0.5) increased from 19.55 km^2^ to 48.34 km^2^ and was mainly distributed in the main center and two sub-centers.

The degrees of urban sprawl on incremental residential land were further calculated using three criteria: (1) residential land expansion is faster than population growth, referring to GR > 0 in Figure 5a; (2) growth type is edge-expansion or leaping, referring to 0 ≤ S < 0.5 in Figure 5b; (3) multi-dimensional sprawl index (SI) is larger than 0.5 in Figure 5c. By an overlay analysis of the three criteria layers, the result of dynamic sprawl assessment was shown in Figure 5d.

During 2000–2010, the area of dynamic sprawl on residential land was 52.30 km^2^ in Hangzhou, accounting for 62.61% of the incremental residential land. For the incremental residential land within the urban core, the average SI is 0.40 and the dominant growth type is infilling, and thus was identified as no dynamic sprawl. For the incremental residential land in the inner suburb, the average level of multi-dimensional sprawl index is 0.48, and the proportions of three growth types were 27% (infilling), 45% (edge-expansion), and 28% (leaping), respectively. There were 27.95% of the total dynamic sprawl in the inner suburb (14.62 km^2^), which were mainly distributed in Xiasha Newtown, Binjiang District, Jiubao, Dingqiao, Sandun, Jiangcun, and Liuxia. For the incremental residential land in the outer suburb, the average level of multi-dimensional sprawl index was as high as 0.57, and those identified as SI > 0.5 were in excess of four-fifths. The leaping type made up about 65% of residential land growth in the outer suburb, and the edge-expansion type accounted for nearly 30%. After all, there were 37.67 km^2^ of residential growth within the outer suburb assessed as dynamic sprawl, three-fifths of which were distributed in Yuhang District (e.g., Linping Newtown, Liangzhu, Yuhang, and Wuchang), and other two-fifths were in Xiaoshan District (e.g., Chengxiang, Shushan, Xintang, Ningwei, Hezhuang, Wenyan, and Daicun). In addition, nearly 5% of incremental residential land cells (1 km × 1 km) saw a certain extent of population shrinkage, which elevated the amount of land consumption per person and resulted in a low-density pattern.

### 3.2. Influencing Factors for the Spatial Variation of SI

Since the GeoDetector method requires categorical variables, the continuous variables were reclassified using the natural break method in ArcGIS 10.2. The Influencing factors were then calculated by factor detector, as shown in Figure 6. Among the 6 variables, the Qiantang River (X2) had not passed the significance test (*p* > 0.05), while others exerted a significant effect on SI changes. 

According to the *q* values in 2000, population growth was the most important factor that affected the spatial distribution of SI, as *q*(X3) = 0.378. Close behind were two land price variables, namely benchmark land price (*q* = 0.335) and land transaction price (*q* = 0.282). Last but not least, planning policies (X6) and the distance to West Lake (X1) also had considerable influences on SI, as *q*(X6) = 0.113 and *q*(X1) = 0.196.

The change of *q* values from 2000 to 2010 indicates that the effect of planning policies was strengthening, while locational and economic factors remained their significant impacts on the spatial variation of SI. During the decade, the explanatory power of the factors, distance to West Lake, population growth, benchmark land price, and land transaction price was greatly lessened. For example, the *q* values of population growth and benchmark land price decreased 0.135 and 0.07, respectively. On the contrary, the variable of urban plan saw an increase of its *q* value from 0.113 to 0.169.

### 3.3. Interaction between Factors for Residential Sprawl

The interactions between these explanatory variables for the variation of sprawl index on residential land were calculated by the interaction detector. Figure 7 shows that the *q*-statistics for factor interactions were higher than the *q*-statistics for every single factor, indicating that the explanatory power of a single factor could be mutually enhanced when interacting with another factor. However, the *q*-statistics for interaction between two factors were less than the sum of the *q*-statistics of two single factors, which suggested that all interactions belong to the bivariate enhancement interaction. 

Population growth not only played a dominant role in the single factor analysis but also has a strong interaction effect with other factors. The dominant interaction between population growth and benchmark land price had the highest *q* values, as *q*(X3∩X4) was 0.546 in 2000 and 0.378 in 2010. Although the *q* value of planning policies (X6) for the spatial variation in SI was lower than the *q* values of population growth (X3), benchmark land price (X4) and land transaction price (X5), the *q* value of their interactions were marvelously higher. For example, the *q* value of the interaction between urban plan and population growth is as high as 0.431 in Figure 7a, and even ranked as the second-largest *q* value in Figure 7b, where *q*(X6∩X3) = 0.338. Similarly, the interaction between urban plan policies and benchmark land price also increased the explanatory power of each factor significantly.

## 4. Discussion

This study investigated the degrees and distribution of residential sprawl in Hangzhou City and assessed the influencing factors for its spatiotemporal variation. The environment-health costs of residential sprawl, effects of urbanization, land market and incremental planning, and policy implications will be discussed in this section. 

### 4.1. Environment-Health Costs of Residential Sprawl

Though sprawl has both negative and positive effects [15,25], the negative impacts outweigh the positive ones [28]. The car-dependent lifestyle and low walking accessibility should be blamed for the rising environment–health costs [18,22]. Scattered development and/or homogenous land-use pattern result in the increasing demand for automobile use [25,28], and thus arouse the issues of transport CO_2_ emissions, obesity, and chronic disease [18,21]. As an example, the correlation between residential sprawl and CO_2_ emissions would be discussed. The number of 1 km × 1 km residential land cells with SI > 0.5 (Table 4) and CO_2_ emissions (Figure 8) differ among different urban districts in Hangzhou. From 2000 to 2010, most districts saw a great increase of residential cells evaluated as SI > 0.5, except Shangcheng and Xiacheng. Districts of Xiaoshan and Yuhang had the greatest growth of residential sprawl, followed by Jianggan. Simultaneously, the increase of CO_2_ emissions was far more intense in Yuhang District (from 3.96 million tons to 12.80 million tons) and Xiaoshan District (from 5.54 million tons to 16.58 million tons), at a growth rate of 2.23 and 2.0 respectively. CO_2_ emissions grew most slowly in Shangcheng (0.77) and Xiacheng (0.77), corresponding to the declining residential sprawl there. It indicates the positive correlation between the spatiotemporal evolution of residential sprawl and CO_2_ emissions, which echoes other research findings [17,18].

### 4.2. Effects of Urbanization, Land Market and Incremental Planning

Being widely regarded as a result of spontaneous market forces [9], urban sprawl in the West is widely regarded as being induced by specific technological innovations like the automobile [30], consumer preference for low-density single-family housing, and government policies like single-use zoning or fragmented local governance [25,31]. Whereas China has been undergoing a transition from state socialism toward a market-oriented institution [11,40], sprawl in China is significantly influenced by the state [6,8]. 

The important influence of urbanization on the residential sprawl was validated by the high *q* values of population growth (*q* = 0.378, 0.243) in the GeoDetector model. Due to the continuous urbanization and preferential policies for attracting talents, a large number of migrants continuously pour into Hangzhou city and settle down. The rise of the migrant population indirectly affects urban sprawl patterns through their housing choices at the urban fringes [9]. Increasing housing demand and the scarcity of spaces in the city center, therefore, motivate residential land expansion to the suburbs. Besides, the diffusion of the manufacturing sector spurs the employment population to gather around suburban industrial agglomerations [10], which may also accelerate the pace of residential suburbanization.

The effect of the land market has also been demonstrated by the high *q* values of benchmark land price (*q* = 0.335, 0.265) and land transaction price (*q* = 0.282, 0.165). The municipal government of Hangzhou city implemented the land reserve system in 1997 and stipulated that real estate developers must obtain the land use rights from the primary land market via bidding and auction. As noted by Liu et al. [40], Hangzhou’s suburban residential development is closely associated with the dynamics of the local land market, which had been undergoing an unprecedented real estate market boom since 2000. By the year 2010, the average transaction price of residential land parcels in suburban areas was 3,723.42 yuan/m^2^, which was only half the level seen at the urban core. The price gap of residential land stimulates real estate developers to build more houses in suburbs. As shown in Figure 9, the side peaks of residential land prices emerged in several sub-centers and new towns of the city. As a result, the amount of residential land area that real estate developers obtained from suburban areas during 2000–2010 were nearly twice as much as that from the urban center. Residential sprawl is related to land financing [8], rooted in market-oriented land reform and fiscal decentralization [10,11,40]. The local government intends to generate immediate financial returns by devoting large tracts of land and charging land transfer fees from real estate developers [8,9,40], always leading to oversupply land and residential sprawl [6]. 

The effect of planning policies on residential sprawl was highlighted by the increasing *q* value of the urban plan and its significant interactions with population growth and land price variables, as *q*(X6∩X3) was 0.431 and *q*(X6∩X4) was 0.342 in Figure 7a. In the context of an imperfect land market and decentralization process, local governments have become the de facto facilitator to guide market forces through a bunch of policies [6]. Among them, an important top-down growth approach is the proactive planning policies which have exerted a dramatic impact on urban sprawl in China [6,9,42]. Potential land resources in the original urban area were extremely limited in Hangzhou before 1996, which compelled the government to expand the administrative districts under jurisdiction. The 1990s’ Hangzhou City Master Plan was proposed to change the monocentric pattern to a polycentric structure, gradually moving the urban core from West Lake to Qiantang River. In this version of the urban plan, 30% of the planned residential areas were located in two sub-centers, i.e., Xiasha and Binjiang, which promoted the process of residential land suburbanization in this period. In the 2000s, the Hangzhou City Master Plan (2001–2020) formally put forward the development strategy of “one main center, three sub-centers and six clusters”, which resulted in the more and more intense residential development along Qiantang River. This was also demonstrated in the GeoDetector model, as the *q* value of West Lake decreased, while the *q* value of urban plan of sub-centers and new towns increased from 2000 to 2010. Along with the implementation of the last versions of the urban master plan and the adjustment of administrative divisions, three sub-centers (i.e., Xiasha, Linping, and Jiangnan) were gradually built into new urban growth poles. After extended urban districts twice in 1996 and 2001, suburban areas were connected more closely to the urban area, which greatly affected the residential land development in Sandun Town, Xiasha Town, Binjiang District, Xiaoshan District, and Yuhang District. By planning a series of new towns, development zones, and educational parks that were discontinuous from the main center [8,11], residential development was encouraged in suburbs despite the lack of adequate public facilities [40]. As a result, residential sprawl has been stimulated in the inner suburb and outer suburb, characterized by a relatively low degree of mixed-use, low population density, low accessibility, and insufficient facilities. Such features of sprawl are always presented in the early stages of sub-center and new towns, which remain to be improved as the development gets into a more mature stage.

### 4.3. Implications for Containing Residential Sprawl

In response to curb urban sprawl, several policies have been adopted by Chinese central government and local governments, such as farmland preservation, urban planning controls, population growth constraints, and land market regulations [9]. Nevertheless, many are often distorted by considerable impulse inherent in land finance [5,8], and what’s more, the failed implementations of these policies encourage extensive urban sprawl [9]. Given the potential for different urban planning strategies and the corresponding interactions with other factors to influence urban sprawl, this study provides policy implications for alleviating urban sprawl and better planning of urban space. By now, incremental planning is still the mainstream in Chinese cities, which takes land expansion as the dominant path to foster urban development by supplying new construction land in priority developing areas. The study has shown that the dynamic sprawl of residential land growth is dominated by leaping development in suburban areas, especially in new towns and residential agglomerations that were set in the context of the urban master plan and spatial plan. There is a long intermediate stage before the new agglomerations gradually mature and transform the city to a polycentric pattern. The increase of population vitality, infrastructure, and urban services in these new towns and suburban agglomerations is often relatively slow, therefore, top-down polycentric urban strategies may aggravate urban sprawl in the initial stage. Policies to attract more capital into education, health, and infrastructure, rather than packing into housing construction projects, may help curb urban sprawl and build a relatively compact, vibrant, and mixed-use new town.

Moreover, the rising awareness of tapping potential and upgrading the stock land has attracted public attention towards stock urban planning and urban renewal, especially those old residential communities and urban villages. An urban renewal movement took place in Hangzhou in its 1980s urban master plan, aimed at contributing 30% of the city’s housing construction. Since then, the municipality has been carrying out numerous large-scale urban reconstruction projects. Through urban renewal, the inefficient use of land and space with low density or poor accessibility is expected to be activated. This is just a starting point to solve the complicated problem of urban sprawl, however, traditional planning methods and procedures are necessary to be changed, and the supporting mechanism and legal rules should be established along with the stock urban planning strategies.

## 5. Conclusions

Among the aspects of sprawl, the residential aspect is central to its status and potential effects. However, inadequate attention has been paid specifically to urban sprawl brought by residential land expansion. This study applied the approach of multi-dimensional measurement to measure urban sprawl on 1 km × 1 km residential land cells, based on its complex connotation mixing multiple dimensions of morphology, density, composition, and accessibility. It may add to the literature on residential sprawl and enrich multi-dimensional measurements of urban sprawl at the micro-scale. Besides, a dynamic assessment was conducted to characterize the spatiotemporal attributes of residential sprawl, between 2000 and 2010. Furthermore, the article provided an interpretation of the intra-city distribution of residential sprawl, by evaluating the explanatory power of factors for the spatial heterogeneity of sprawl index via the GeoDetector modeling method. This may contribute to a preliminary examination under China’s context of how government and market forces influence the spatiotemporal variation of residential sprawl. 

Hangzhou saw incredible urban sprawl on residential land since 2000. Taking the multi-dimensional sprawl index SI > 0.5 as the criterion of sprawl, the area of residential sprawl tripled in ten years. More than half of residential land were above the average level of SI, characterized by fragmented and scatter land patches, low population density, homogeneous land use, and poor accessibility. Among the incremental residential land, more than three-fifths was evaluated as dynamic sprawl, which satisfied three criteria simultaneously: leaping or edge-expansion, faster land expansion compared to population growth, multi-dimensional sprawl index SI > 0.5. The spatial distribution of residential sprawl changed tremendously over time. Most sprawling residential land cells were distributed in the inner suburb at the beginning, while the residential sprawl in the outer suburb has dominated since 2000. Residential land that was not sprawling was mainly distributed in the main center and two sub-centers. The dynamic sprawl of residential growth was mainly distributed in new towns and residential agglomerations of the inner suburb and outer suburb. Leapfrogging residential development, homogenous land-use pattern, and the low accessibility from one land use to another result in the increasing demand for automobile use, and thus arouse environment-health consequences, which can be substantiated by the positive correlation between residential sprawl and CO_2_ emissions.

The potential effects of urbanization, land market, and planning policy on the spatial variation of residential urban sprawl were verified in GeoDetector modeling. Among them, population growth and land price ranked as the most important factors by their high *q*-statistics, which indicated the significant effects of rapid urbanization and land market on accelerating the residential sprawl. The spatial variation of land price demonstrated a positive corresponding relationship between land price and residential growth. The increasing *q* value of the urban plan and its interaction with other factors highlighted the effects of incremental planning policies. Through proactive planning policies, the local government accelerated the population growth and booming residential land market in suburban areas. The radical cause of land oversupply and residential sprawl could be attributed to the local government’s fervor for land financing. In China’s context of imperfect land market and decentralization, the proactive planning policies of sub-centers and new towns have dramatic impacts on urban sprawl, in a top-down approach. As a result, development in the urban fringe always taken on a non-mixed and homogenous pattern before they get into a more mature stage. Therefore, it is necessary to transform the incremental planning mode, to curb urban sprawl. 

The present study has its limitations. Firstly, the heterogeneity of urban sprawl is embodied both in the intra-city variation and in the inter-city variation, this study only observed the intra-city variation of residential sprawl. The approach proposed in this study needs to be further verified by comparison analysis on the inter-city variation of urban sprawl. Secondly, owing to incomplete data, this study only observed the urban sprawl on residential land at 1 km × 1 km cells in 2000 and 2010, research on a wider temporal scale and different unit scales would be more conducive to improve the understanding of China’s residential sprawl. These need further research in the future. Moreover, continuous measuring of sprawl’s environment-health costs is also one of the key points of future research.

## Figures and Tables

**Figure 1 ijerph-18-08619-f001:**
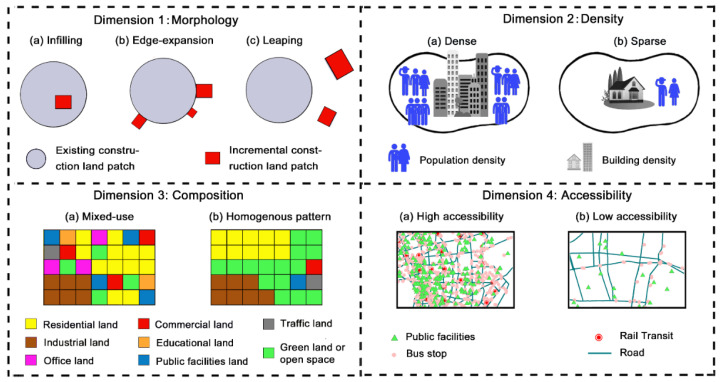
Multiple dimensions of urban sprawl. Note: (**a**) refers to a not sprawl situation, while (**b**,**c**) indicate a relatively sprawling situation.

**Figure 2 ijerph-18-08619-f002:**
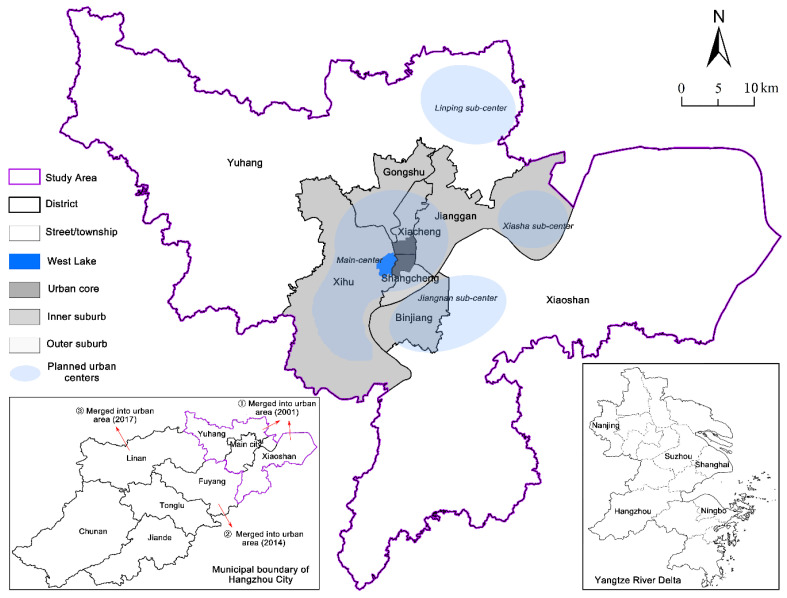
Map of the study area.

**Figure 3 ijerph-18-08619-f003:**
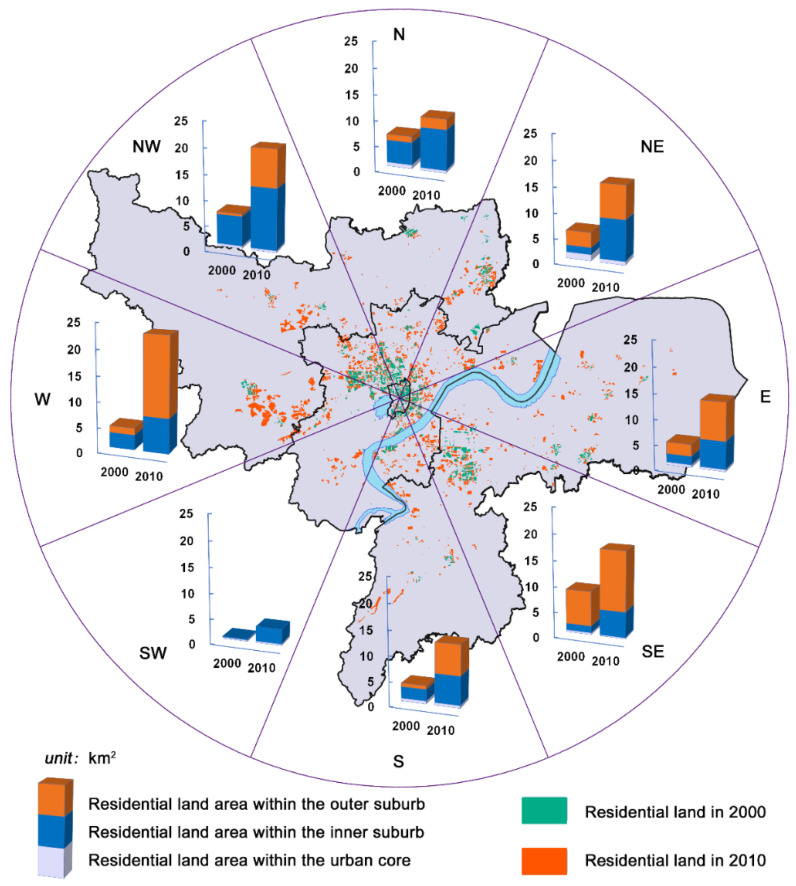
The residential land layout of Hangzhou in 2000 and 2010 (km^2^).

**Figure 4 ijerph-18-08619-f004:**
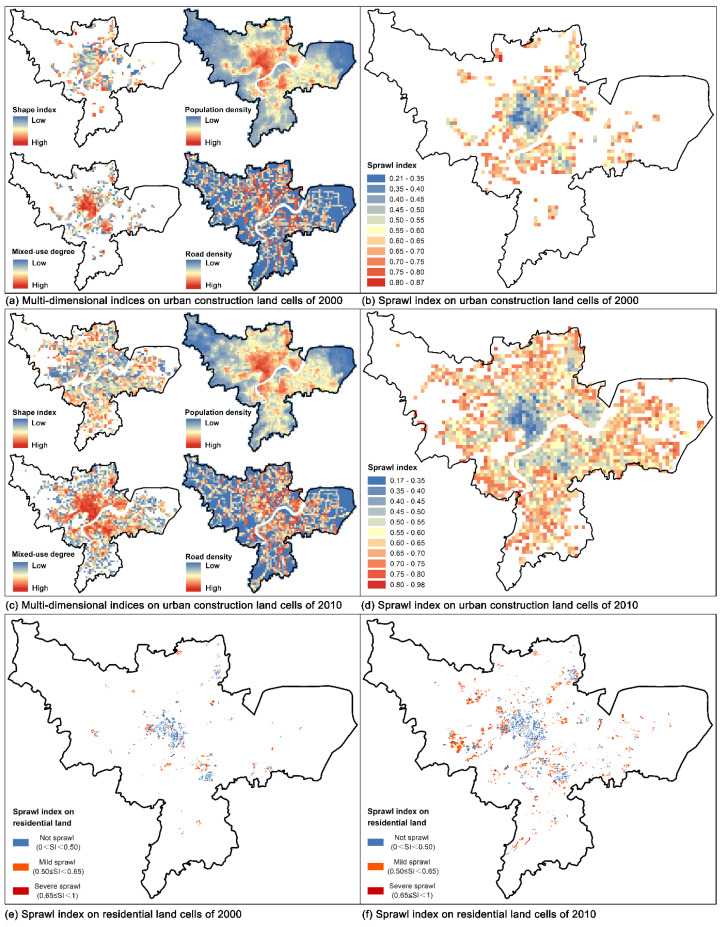
The layout of multi-dimensional measuring indices and sprawl index on 1 km × 1 km cells: (**a**) Multi-dimensional indices on urban construction land cells of 2000; (**b**) Sprawl index on urban construction land cells of 2000; (**c**) Multi-dimensional indices on urban construction land cells of 2010; (**d**) Sprawl index on urban construction land cells of 2010; (**e**) Sprawl index on residential land cells of 2000; (**f**) Sprawl index on residential land cells of 2010.

**Figure 5 ijerph-18-08619-f005:**
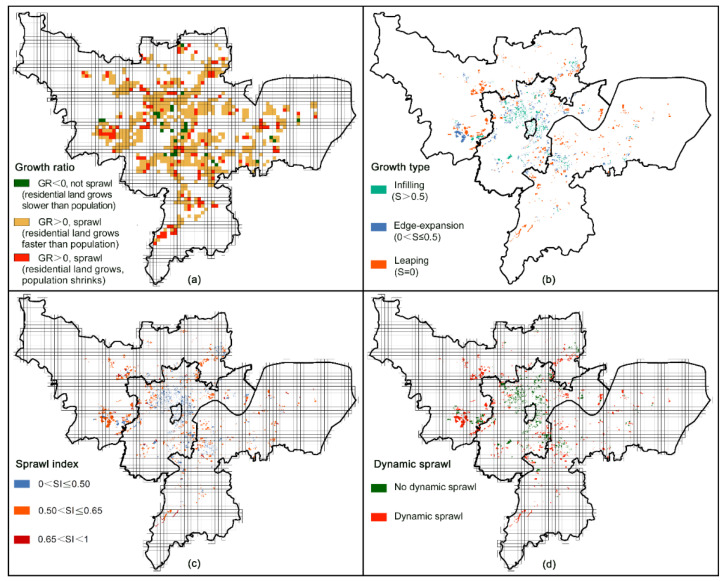
Dynamic sprawl on 1 km × 1 km cells of incremental residential land: (**a**) Growth ratio of residential land growth to population growth; (**b**) Growth types of the incremental residential land patches; (**c**) Multi-dimensional Sprawl index; (**d**) Dynamic assessment result of urban sprawl.

**Figure 6 ijerph-18-08619-f006:**
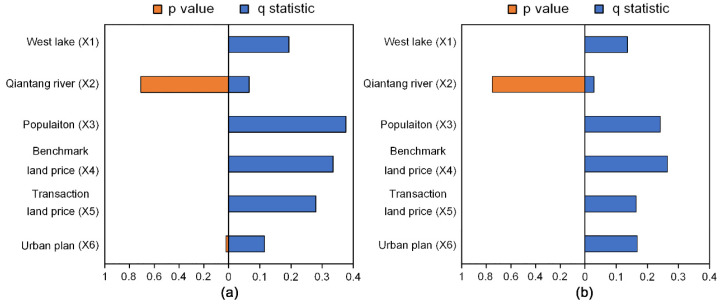
The *q*-statistic and *p* value calculated by factor detector: (**a**) Factor Detector results of residential sprawl in 2000; (**b**) Factor Detector results of residential sprawl in 2010.

**Figure 7 ijerph-18-08619-f007:**
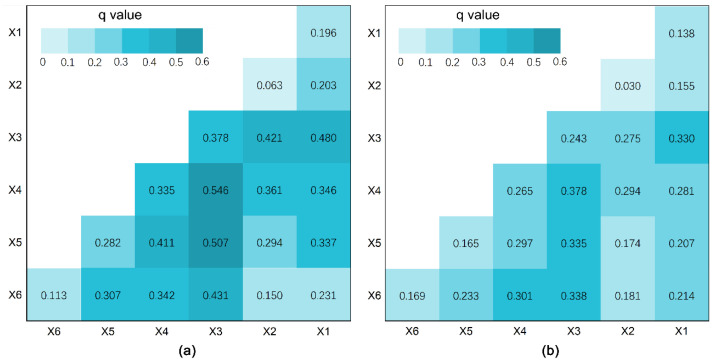
The interactions between two explanatory variables for the variation of sprawl index on residential land: (**a**) The *q*-statistics calculated by interaction detector in 2000; (**b**) The *q*-statistics calculated by interaction detector in 2010.

**Figure 8 ijerph-18-08619-f008:**
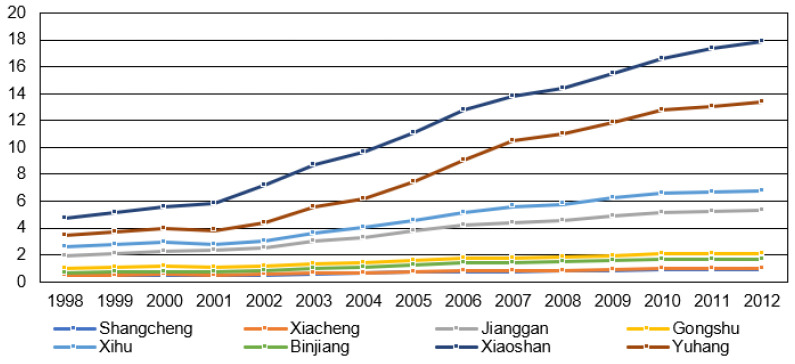
CO_2_ emissions in each district from 1998 to 2012 (million tons).

**Figure 9 ijerph-18-08619-f009:**
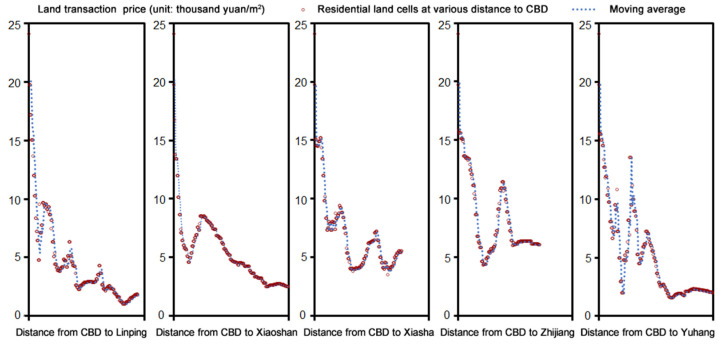
Variations of Hangzhou’s residential land prices from the urban center to sub-centers during 2000–2010.

**Table 1 ijerph-18-08619-t001:** Types of urban sprawl measurements in the literature.

Type	Literature	Indicator	Scale
Single Dimension	Fulton et al. [31]	density	macro-scale; dynamic
	Lopez and Hynes [14]	density	macro-scale; static
	Gao et al. [27]	growth ratio of land to population	meso-scale; dynamic
Multiple Dimensions	Galster et al. [26]	density, nuclearity, centrality, concentration, proximity, and clustering; other two dimensions, continuity, and heterogeneity, were not measured	meso-scale; static
	Jat et al. [34]	Shannon’s entropy, landscape metrics (i.e., patchiness and map density)	meso-scale; dynamic
	Song and Knaap [35]	connectivity, density, accessibility, pedestrian access, land-use mix.	micro-scale; dynamic
	Frenkel and Ashkenazi [29]	density, scatter, land-use composition	meso-scale; dynamic
	Hamidi and Ewing [30]	density, land-use mix, activity centring, and street accessibility	meso-scale; dynamic
	Zeng et al. [36]	composition, configuration, proximity, accessibility, gradient, density, and dynamics	multi-level scales; dynamic
	Triantakonstantis and Stathakis [37]	population density, landscape metrics (i.e., shape, aggregation, compactness, and dispersion)	macro-scale; dynamic

**Table 2 ijerph-18-08619-t002:** The Multi-dimensional Indices of Urban Sprawl.

Dimensions	Indices	Description	Direction
Morphology	Shape index	The shape index of built-up area is calculated by the formula [29]: $Shape=\frac{L_{i}^{c}}{A_{i}^{c}}$$,\;\mathrm{where}\;L_{i}^{c}$$\;\mathrm{and}\;A_{i}^{c}$ refer to the perimeter and area of urban construction land in unit *i*.	+
Density	Population density	The population density is calculated as the number of people per square kilometer by using the approach of random Forest-based dasymetric redistribution (Data source: www.worldpop.org).	−
Composition	Mixed-use degree	The mixed-use degree of urban land is calculated by using a modified entropy index [43]: $Mix=\frac{-\sum_{k=1}^{K_{i}}P_{k,i}\ln P_{k,i}}{\ln K_{i}}$, where $K_{i}$ is the number of land-use categories, $P_{k,i}$ refers to the proportion of land-use category *k* in unit *i* to the total area of unit *i*. The value of MIX ranges between 0 (homogeneity, a single land-use) and 1 (heterogeneity, evenly distributed among all land-use categories).	−
Accessibility	Road density	Road accessibility is calculated by the formula: $Road=\frac{A_{i}^{r}}{A_{i}\;}$$,\;\mathrm{where}\;A_{i}$ is the total area of unit *i*, $A_{i}^{r}$ refers to the road area in unit *i* $(A_{i}^{r}=L_{i}^{r}w_{i}^{r}$$,\;\mathrm{here}\;L_{i}^{r}$$\;\mathrm{is}\;\mathrm{road}\;\mathrm{length}\;\mathrm{in}\;\mathrm{unit}\;i,\;w_{i}^{r}$ is road width, which varies among different road grades).	−

**Table 3 ijerph-18-08619-t003:** Variables for GeoDetector modeling.

Types	Factor	Description
Dependent variable	Urban sprawl index	Result of the multi-dimensional measurement
Independent variables	West lake (*X1*)	Distance to West Lake
	Qiantang river (*X2*)	Distance to Qiantang River
	Population growth (*X3*)	Population growth during 1990–2000 and 2000–2010, derived from population census and WorldPop website (www.worldpop.org)
	Benchmark land price (*X4*)	Benchmark land price approved by local government
	Transaction land price (*X5*)	Residential land price in the transaction crawled from the website (land.fang.com)
	Urban plan (*X6*)	Distance to the main center, sub-centers, and new towns, drawn by the authors

**Table 4 ijerph-18-08619-t004:** Descriptive statistics of sprawl index on the residential land of each district.

Districts	2000		2010	
	Mean SI	Cells with SI > 0.5	Mean SI	Cells with SI > 0.5
Shangcheng	0.519	26	0.498	15
Xiacheng	0.425	5	0.401	4
Jianggan	0.498	27	0.514	54
Gongshu	0.465	10	0.484	16
Xihu	0.559	41	0.549	68
Binjiang	0.615	23	0.544	40
Xiaoshan	0.609	86	0.590	241
Yuhang	0.612	68	0.596	188

## Data Availability

The data presented in this study are available on request from the corresponding author.

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
