# Peer review of "Evaluating Spatiotemporal Distribution of Residential Sprawl and Influencing Factors Based on Multi-Dimensional Measurement and GeoDetector Modelling"

_ijerph, 2021, doi:10.3390/ijerph18168619_

Round 1

Reviewer 1 Report

The topic of land consumption due to urban sprawl and its impacts in term of environmental, social and economic issues is relevant and up-to-date.

The methodology is sound and adequately presented.

The conclusion could benefit from a more detailed description of the results and follow up. 

Author Response

Dear reviewers,

We appreciate the kind and constructive comments from you. The manuscript has been revised with a tracked and a clean version attached.

Response to Reviewer 1 Comments

Point 1: The topic of land consumption due to urban sprawl and its impacts in term of environmental, social and economic issues is relevant and up-to-date. The methodology is sound and adequately presented. The conclusion could benefit from a more detailed description of the results and follow up.

Response 1: We appreciate your constructive comments. More detailed descriptions of the results were enriched in the conclusion section. As another reviewer made the same comment and suggested to “remove paragraphs from Results and Discussion and put them in the Conclusion”, we made a big change to better organize the three sections. We adjusted the content in original section 3.1 into the new section 3.1 “Multi-dimensional sprawl index on residential land cells” and new section 4.1 “Environment-health costs of residential sprawl”. The paragraphs in original section 4.3 “Validations and limitations of this study” were put in the conclusion section.

Besides, as judged by the reviewer, English language and style are fine/minor spell check required, the words and sentences were checked by a native English-speaking colleague. 

Reviewer 2 Report

The manuscript titled "Evaluating Spatiotemporal Distribution of Residential Sprawl and Influencing Factors Based on Multi-Dimensional Measurement and GeoDetector Modelling" intends to build a multi-dimensional measurement approach, and an empirical case study conducted to measure the temporal and spatial dynamic characteristics of residential sprawl in Hangzhou City. The authors use the method of GeoDetector modelling and examine the influencing factors for the variation of residential sprawl index quantitatively. The empirical basis of the paper is a qualitative research study, including a GeoDetector model which constructed with a GIS. Also, include direct and indirect, standardized indicators based on its complex connotation mixing multiple dimensions of morphology, density, composition, and accessibility.

The research is original; it could be characterized as novel and in my opinion important to the field, it also has the appropriate structure and language been used well. In the meanwhile, the manuscript has a nice extent (about 6,900 words), the tables (3) and figures (9) make the paper to reflect well to the reader. For this reason, paper has a "diversity look", not only tables, not only numbers, not only words.

The title is all right. The abstract reflects well the findings of this study, but it has a long length (about 285 words). The introduction is effective, clear, and well organized; it really introduced and put into perspective what research is negotiating.

Please, revise the abstract, it must be up to 200 words long, for this reason I would be good to reduce [see: Instructions for Authors / Manuscript Preparation / Front Matter / Abstract: - (https://www.mdpi.com/journal/ijerph/instructions#preparation or https://www.mdpi.com/files/word-templates/ijerph-template.dot)].

For the Methodology chapter, the research conduct has been tested in several areas of the world, with similar results and will probably be tested in others. In this way it is documented that a project which is tested in a place with its own characteristics can be implemented in other places around the world. About the references of the manuscript the authors include references which are already exists in bibliography from the entire world (Asia, America, Europe, and Australia e.tc.). Moreover, I would appreciate it if also saying more about the PQI (a paragraph or two I think it will be all right).

The results and discussion sections are very good. The argument flows and is reinforced through the justification of the way elements are interpreted. The same applies to the conclusions, which it could be longer.

It is advised to revise the Discussion and Conclusion. Both sections should be consistent in terms of Proposal, Problem statement, Results, and of course, future work. Your conclusion section is too short and does not do justice to your work. Make it your key contributions, arguments, and findings clearer. You must refer to the literature and previous studies in your discussion and conclusion sections. It is recommended to remove paragraphs from Results and Discussion and put them in the Conclusion, with nice order and to be enriched.

Please, revise the references, they must have an appropriate style, for this reason I would be good to reduce [see: Instructions for Authors / Manuscript Preparation / Back Matter / References: - (https://www.mdpi.com/journal/ijerph/instructions#preparation or https://www.mdpi.com/authors/references)].

Author Response

Dear reviewers,

We appreciate the kind and constructive comments from you. The manuscript has been revised with a tracked and a clean version attached.

Point 1: Please revise the abstract, it must be up to 200 words long, for this reason I would be good to reduce.

Response 1: As suggested by the reviewer, we simplified the abstract from 285 words to 216 words, we hope it would be acceptable given the common excess situations occurred in latest articles on IJERPH.

Point 2: For the Methodology chapter, the research conduct has been tested in several areas of the world, with similar results and will probably be tested in others. In this way it is documented that a project which is tested in a place with its own characteristics can be implemented in other places around the world. About the references of the manuscript the authors include references which are already exists in bibliography from the entire world (Asia, America, Europe, and Australia e.tc.). Moreover, I would appreciate it if also saying more about the PQI (a paragraph or two I think it will be all right).

Response 2: The authors are confused about the acronym “PQI” (we are sincerely sorry that we could not figure out what does it mean after searching literature using the keyword “PQI”). As the reviewer mentioned the methodology chapter and references, we tried our best to revise the methodology section 2.1.1 as below:

“On the basis of the literature, urban sprawl is a complex phenomenon that cannot be measured by only one or two measures [29]. Instead, the multi-dimensional connotation of sprawl requires various of measures for each dimension [13,26]. Scholars have asserted that characterizations of sprawl are not similar across the world [26,28,35], which poses a challenge for the generalizability of measurements. Seeking to account for the unique characteristics of regions, a few multi-dimensional measurements even confounded the causes and consequences of urban sprawl with the phenomenon itself, which was widely criticized [13,15,26]. Findings from different multi-dimensional measurements usually cannot be compared with each other and, therefore, may be difficult to interpret consistently [15]. After all, an ideal measuring approach should be applicable as well as duplicable nationally or internationally [14]. Based on the principle, this research proposed a multi-dimensional measuring approach that capture the cardinal dimensions of sprawl: morphological discontinuity, low density, low mixed-use degree of land-use composition, and poor accessibility. The details of the four dimensions were already described above, referring to Figure 1. A complete ex-planation of all indices used in this study and the direction of their impact on the degree of sprawl are presented in Table 2.”

Point 3: The results and discussion sections are very good. The argument flows and is reinforced through the justification of the way elements are interpreted. The same applies to the conclusions, which could be longer. It is advised to revise the Discussion and Conclusion. Both sections should be consistent in terms of Proposal, Problem statement, Results, and of course, future work. Your conclusion section is too short and does not do justice to your work. Make it your key contributions, arguments, and findings clearer. You must refer to the literature and previous studies in your discussion and conclusion sections. It is recommended to remove paragraphs from Results and Discussion and put them in the Conclusion, with nice order and to be enriched.

Response 3: We appreciate your constructive comments.  To better organize the results and discussion section, we adjusted the content in original section 3.1 into the new section 3.1 and new section 4.1. The paragraphs in original section 4.3 on the validations and limitations of this study were put in the conclusion section and at the head paragraph of discussion section. Given that the length of conclusion section in most IJERPH published papers are kept simple and short, we responded to your meaningful suggestion by enriching some detail descriptions of the results in the conclusion section rather than to enlarge this section too much. Furthermore, we referred to the literature and previous studies in the discussion section, including section 4.1, 4.2, and 4.3.  

For instance, “Though sprawl has both negative and positive effects[15,25], the negative impacts outweigh the positive ones[28]. The car-dependent lifestyle and low walking accessibility should be blamed for rising the environment-health costs[18,22]. Scattered development and/or homogenous land-use pattern result in the increasing demand for automobile use[25,28], and thus arouse the issues of transport CO2 emissions, obesity and chronic disease[18,21]. …… It indicates the positive correlation between the spatiotemporal evolution of residential sprawl and CO2 emissions, which echoes other re-search findings[17,18].” in section 4.1;

“Being widely regarded as a result from spontaneous market forces[9], urban sprawl in the West is widely regarded as being induced by specific technological innovations like the automobile[30], consumer preference for low-density single-family housing, and government policies like single-use zoning or fragmented local governance [25,31]. Whereas China has been undergoing transition from state socialism toward a market-oriented institution[11,40], sprawl in China is significantly influenced by the state[6,8].……The rise of migrant population indirectly affects urban sprawl pattern through their housing choices at the urban fringes[9]. ……Besides, the diffusion of manufacturing sector spurs employment population to gather around suburban industrial agglomerations[10]……As noted by Liu et al. [40], Hangzhou’s suburban residential development is closely associated with the dynamics of the local land market……Residential sprawl is related to land financing[8], rooted in the market-oriented land reform and fiscal decentralization[10,11,40]. The local government intends to generate immediate financial returns by devoting large tracts of land for low-density residential communities and charging land transfer fee from real estate developers [8,9,40], always leading to oversupply land and residential sprawl[6]. …… In the context of an imperfect land market and decentralization process, local governments have become the de facto facilitator to guide market forces through a bunch of policies[6]. Among them, an important top-down growth approach is the proactive planning policies which has exerted dramatic impact on urban sprawl in China[6,9,42].……By planning a series of new towns, development zones and educational parks that were discontinuous from the main center[8,11], residential development was encouraged in suburbs despite the lack of adequate public facilities[40].” in section 4.2;

“In response to curb urban sprawl, several policies have been adopted by central government and local governments, such as farmland preservation, urban planning controls, population growth constraints, and land market regulations[9]. Nevertheless, many are often distorted by considerable impulse inherent in land finance[5,8], and what’s more, the failed implementations of these policies encourage extensive urban sprawl[9].”in section 4.3;

Point 4: Please, revise the references, they must have an appropriate style, for this reason I would be good to reduce.

Response 4: As suggested by the reviewer, we carefully reduced the amount of references from 59 to 47 and reedited the style by using MDPI.ens references style file from the EndNote website.

Reviewer 3 Report

This study addresses the issue of urban sprawl in a case study of Hangzhou city. It purposes to fill the gaps of measuring the residential sprawl and interpreting the influencing factors.

The manuscript is well organized, the reader easily reads the text, tables and figures; the topic is strategic for territorial planning. Hence, the paper can be judged to add knowledge in its topic and it adheres to the “IJERPH” standards.

Only small questions:

1) Line 218 “From the West Lake era to the Qiantang River era”, what does “era” mean?

2) Expand the topic of environment-health costs of urban sprawl in the "Discussion" session.

Author Response

We appreciate the kind and constructive comments from you. The manuscript has been revised with a tracked and a clean version attached.

Point 1:  Line 218 “From the West Lake era to the Qiantang River era”, what does “era” mean?

Response 1:  Here, “era” is a phrase similar as “period”, “phase” or “stage”, it refers to significant strategic stage of urban plan and city construction. For a long time in history, all the urban districts of Hangzhou were situated on the north bank of Qiantang River, when was called the era of West Lake because urban construction spread outward with West Lake as the center. In order to get more space for urban development, the city government put forward the strategic transition “From the West Lake era to the Qiantang River era”, which comprises a series of spatial policies and administrative division adjustment. Binjiang District and Xiaoshan District were merged into urban districts of Hangzhou on 1996 and 2010 successively, which formed a new city center on the south bank of Qiantang River, and that marked a new phase in the urban construction history of Hangzhou city, in other words, the era of Qiantang River began. For more detail descriptions about the strategy, the reviewer can also refer to the section 4.2.

Point 2:  Expand the topic of environment-health costs of urban sprawl in the "Discussion" session.

Response 2: We appreciate your constructive suggestion, and a new section 4.1 was added to discuss the environment and health costs of residential sprawl:

“Though sprawl has both negative and positive effects [15,25], the negative impacts outweigh the positive ones [28]. The car-dependent lifestyle and low walking accessibility should be blamed for rising the environment-health costs [18,22]. Scattered development and/or homogenous land-use pattern result in the increasing demand for automobile use [25,28], and thus arouse the issues of transport CO2 emissions, obesity and chronic disease [18,21]. As an example, the correlation between residential sprawl and CO2 emissions would be discussed. The number of 1km×1km residential land cells with SI>0.5 (Table 4) and CO2 emissions (Figure 8) differ among different urban districts in Hangzhou. From 2000 to 2010, most districts saw a great increase of residential cells evaluated as SI>0.5, except Shangcheng and Xiacheng. Districts of Xiaoshan and Yuhang had the greatest growth of residential sprawl, followed by Jianggan. Simultaneously, the increase of CO2 emissions is far more intense in Yuhang District (from 3.96 million tons to 12.80 million tons) and Xiaoshan District (from 5.54 million tons to 16.58 million tons), at a growth rate of 2.23 and 2.0 respectively. CO2 emissions grew most slowly in Shangcheng (0.77) and Xiacheng (0.77), corresponding to the declining residential sprawl in there. It indicates the positive correlation between the spatiotemporal evolution of residential sprawl and CO2 emissions, which echoes other research findings [17,18].”

This manuscript is a resubmission of an earlier submission. The following is a list of the peer review reports and author responses from that submission.

Round 1

Reviewer 1 Report

This paper is of interest to its object but must be completely revised. The introduction should highlight its interest to the international reader. In the literature review, more citations should be used. The work of Hoggart and Paniagua (2001) in the Journal of rural studies is interesting on the processes of restructuring and the role of the state and civil society. A seminal reference is P. Cloke The rural state?. London, Clarendon, 1990.  The methodology must justify the choice of the case study, its relevance in the international and national context. In the analysis of results, it is necessary to justify the choice of variables in the context of the sources of information used. The conclusions must have international relevance. The references are insufficient. The paper needs more work.

Reviewer 2 Report

I recognise that there is a lack of research into the rapid suburbanization experienced by developing countries and need for better understanding of the interplay between the various stakeholders including local government planners, property developers, real estate investors and residents.  However the article goes about the task in a disjointed and (at times) incoherent way. The article is too long and comes across as two separate and incompatible pieces of research that have been combined together - one an empirical and statistical analysis of characteristics of place in relation to land prices and the other an interpretivist consideration of the interplay between residential market and stakeholders.  I am afraid that this just does not work.

Fundamentally the article fails to recognise some of the core and underpinning theory specifically urban growth, land use and bid-rent theory, the latter of which was only mentioned briefly towards the end of the article. There is also only partial recognition of stakeholders and actors in the real estate development process when there is lots of literation and models of the way the stakeholders interact within prevailing market conditions and governance  arrangements particularly in respect of China's state controlled planning, development rights and land markets.  The article also fails to clearly define the topic in terms of what suburban development is, frequently conflating new towns and development zones; no mention is made of urban extensions, infill or peri-urban development. 

The researchers approach is also confused with early mention made of interpretive framework and research questions but followed by quantitative methods and statistical analysis more suited to hypothesis testing.  

There was a general absence of understanding of supply and demand theory and basic property economics from the perspective of housing land supply and concepts of inelasticity.  First mention of supply and demand in section 3.2.2 of the article.

The data used for the study covers a ten year period between 2000 and 2010 which means that it is already over a decade out of date.  The analysis of explanatory variables needs to be more clearly founded on recognised theory of residential development land markets.

The sequence of the presentation of content was also peculiar with a lot of the issues and content in the discussion section being needed to be covered in the literature review and context setting. e.g. first mention of urban growth theory on page 11, much of what follows seemed superfluous to requirements or part of a different article. It was only on page 12 that recognition was given to the importance of transport and other infrastructure to support residential extensions and the influence of transport networks. Fundamentally the entire premise of the article is undermined by the acknowledgement in section 5.4 that local governments completely monopolize the supply of (development) land. The article then starts straying into issues of cross subsidy and betterment.    

The conclusion are brief and lack validity.

A general point of concern was the frequent and repeated absence of reference for much of the content which is clearly not original material.

A suggestion to the authors would be to separate the content of the article into two separate articles, using quantitative analysis of more up to date data to test hypotheses around characteristics of development land for urban extensions in China, to be soundly based on internationally recognised and relevant theory; and a second article exploring stakeholder and governance theory and models to explore the influence, control and interplay between these various actors within the context of a state controlled planning and development in respect of urban extensions in China.  Careful thought would be needed about the two different types of journals to which any future articles should be submitted. 

Reviewer 3 Report

The article deals with a very interesting and important issue. This problem exists in many countries and therefore the text may be of interest to many readers. However, in my opinion, it requires major improvements:
1. in the abstract, please elaborate on what new contribution the article gives;
2. a large international literature review on suburbanization, the real estate market and the role of public authorities (in the context of different concepts, e.g. Governance) is definitely missing. In my opinion, it is necessary to add a very extended section on these topics and to align the rest of the article with them. Without this international context, it will be more difficult to publish the article.
3. when describing Chinese cases, I suggest to take a more problematic view - when describing specific situations (e.g. expropriation of rural land by authorities, it is worth embedding this in the international literature on the subject);
4) I propose to pay much more attention to the problems and risks that are associated with suburbanization. there are many of them;
5. i also propose to refer in more depth to the framework and opportunities provided by the Chinese land use system;
6. from the discussion, move to other parts of the article those passages that are simple characterisations of spatial policy tools or development policies;
7. the discussion must include direct reference to the literature - both that cited in the review and others. It cannot be limited to general conclusions.

I think it will not be a problem for the authors to take these comments into account. The topic is very interesting and, if corrected, the article may be of great interest.